# A mode-locked random laser generating transform-limited optical pulses

**Jean Pierre von der Weid** [1], **Marlon M. Correia**[1], **Pedro Tovar** [2], **Anderson S. L. Gomes**[3] & **Walter Margulis** [1,4] ✉

Ever since the mid-1960's, locking the phases of modes enabled the generation of laser pulses of duration limited only by the uncertainty principle, opening the field of ultrafast science. In contrast to conventional lasers, mode spacing in random lasers is ill-defined because optical feedback comes from scattering centres at random positions, making it hard to use mode locking in transform limited pulse generation. Here the generation of sub-nanosecond transform-limited pulses from a mode-locked random fibre laser is reported. Rayleigh backscattering from decimetre-long sections of telecom fibre serves as laser feedback, providing narrow spectral selectivity to the Fourier limit. The laser is adjustable in pulse duration (0.34–20 ns), repetition rate (0.714–1.22 MHz) and can be temperature tuned. The high spectral-efficiency pulses are applied in distributed temperature sensing with 9.0 cm and $3.3 \times 10^{-3}$ K resolution, exemplifying how the results can drive advances in the fields of spectroscopy, telecommunications, and sensing.

Laser light differs from spontaneous emission owing to the existence of cavity modes with well-defined and equally spaced frequencies. A mode-locked laser can generate transform-limited pulses when all modes are perfectly phase-locked to each other through a coupling mechanism such as periodic phase or amplitude modulation[1–4]. Random lasers[5–8] are based on a stochastic distribution of scattering centres that help form an effective cavity with laser modes, even when a laser mirror is lacking[9–11]. The random mode-spacing makes it difficult to lock the phases of the modes to each other[12–21]. Mode-locking random lasers where the gain and scattering coexist spatially, as in dyes[7] or semiconductor powders[22], is even more challenging. For this reason, few random laser reports exist on locking spatial modes[11–20] or the longitudinal modal distribution[21,23], and none to the Fourier transform limit. Stable picosecond pulse generation has also been reported in a random fibre laser exploiting Raman gain[24].

The generation of transform-limited pulses in random lasers can be attempted in a simplified system. In random single-mode fibre lasers[25], the complexity is reduced to one spatial dimension. Furthermore, it is possible to use lumped amplifiers to separate the section providing gain from the section with randomly distributed scatterers[26–28]. Random

feedback from a Telecom fibre lies at the heart of the mode-locked transform-limited pulse generation discussed here. The mean back-scattered power guided back in standard single-mode fibres amounts to a fraction −72 dB/m of the forward propagating power. The extremely faint Rayleigh backscattering in a short section of fibre (3 cm–2 m) arbitrarily selected within a much longer fibre spool (10's m–100's m) serves as laser feedback. The chosen backscattering section of fibre is much shorter than the entire laser cavity. Periodically gated optical amplification allows synchronising the arrival of the backscattered light from the distributed mirror to the opening of the gain window. Light scattered either too early or too late (from points lying before or after the chosen "addressed" section) does not experience periodic amplification and dies off. Provided the amplification in a roundtrip compensates for the incurred loss, the laser reaches threshold. The duration of the optical pulse defines the length of the section, which acts as a distributed laser mirror.

Similar to all other fibre segments, this section has an unknown and random distribution of scattering centres. However, the coherent sum of all backscattered optical fields from this segment at a given wavelength has a fixed (albeit unpredictable) value if external

[1]Centre for Telecommunication Studies, Pontifical Catholic University of Rio de Janeiro, Rio de Janeiro 22451-900 RJ, Brazil. [2]Department of Physics, University of Ottawa, Ottawa K1N 6N5 Ontario, Canada. [3]Departamento de Física, Universidade Federal de Pernambuco, Recife 50670-901 PE, Brazil. [4]Department of Applied Physics, Royal Institute of Technology, Roslagstullsbacken 21, 106 91 Stockholm, Sweden. ✉e-mail: Margulis@kth.se

conditions such as temperature and strain are constant. The integrated phase value of the extended random mirror depends critically on the wavelength (optical frequency) of the input beam[29]. While some wavelengths experience mainly destructive interference, others add constructively to a peak in reflectivity, as commonly observed in phase optical time-domain reflectometry (Φ-OTDR)[30,31]. This has been referred to as the Rayleigh fingerprint[32] of the fibre section. Therefore, the chosen backscattering section of passive fibre behaves at its entrance as a time-constant faint coloured mirror reflecting various spectral peaks, like those obtained by printing low reflectivity random Bragg gratings[26]. The Rayleigh scattered light guided back from the extended random mirror repetitively feeds the laser cavity with a constant phase signal at the dominant wavelength, and spectrally filters laser emission. With periodic amplification, the highest of the reflected spectral peaks dominates, imprinting its central wavelength and bandwidth on the laser pulse formed. This backscattered signal can be reproduced after a phase-shift equal to an integral number of 2π radians accumulated along a roundtrip, and the modes of the laser cavity can be locked.

In this work, synchronous mode-locking[33] is achieved with repetitive amplification of the circulating signal, where the pulse duration is largely determined by the time-gated amplification and by the sharply filtered feedback[34]. The pulses generated by the laser are found to be exactly or nearly transform-limited in a range of pulse durations (0.34–20 ns) and cavity lengths (170–285 m). When choosing the laser pulse duration, one also automatically chooses the length of the spectral filter and its bandwidth, thus maintaining the time-bandwidth limit. Remarkably, a few centimetres of highly transparent single-mode telecom fibre can be used as a mirror in a laser cavity, one with narrow linewidth peaks that guarantee transform-limited pulse generation.

## Results
### Theory and simulations

Rayleigh scattering in optical fibres is generated by random fluctuations of the refractive index at the nanoscopic scale, but the interference of Rayleigh scattered coherent light and the backscattered intensity are governed by random fluctuations of the mean refractive index on a much larger scale. Backscattering of coherent light can be modelled as the sum of the contributions from every portion of the fibre in which the phase of each contribution is randomized[29]. This allows evaluating the reflectivity of the distributed random mirror embodied by a chosen section of the backscattering fibre. The intensity spectrum of the backscattered light at the input point is frequency-dependent and fluctuations can be described as:

$$I(k) \propto |\sum_{i=1}^{N} \exp(j\bar{n}_i kd)|^2 = \langle I \rangle + \sum_{i \neq p} \exp(j[\bar{n}_i - \bar{n}_p]kd) \qquad (1)$$

where the fibre section is considered as a series of $N$ segments of mean length $d$ and mean refractive index $\bar{n}_i$ with random variations from segment to segment. The second term of Eq. (1) is a double sum over all pairs of segments $i$ and $p$ and describe the fluctuations due to the interference between the optical field from all segments. The result is an intensity spectrum with strong frequency dependence, with fluctuations that can reach 10 dB above the mean value in kilometre-long fibres[29], coming from constructive interference of the backscattered fields. As mentioned above, although extremely faint, this backscattered light can trigger laser action in a random fibre laser, provided there is sufficient gain in the roundtrip. Previously, kilometre lengths were used with Raman[35] and Brillouin[36] amplification to build random fibre lasers. Scattering in long fibre lengths was also used in combination with lumped amplification such as in SOA- or EDFA-based random lasers[11,37].

The backscattered intensity spectrum calculated using Eq. (1) for typical standard single-mode fibres and $d$ ~ 5 mm is shown in Fig. 1a–c within a 10-GHz range of optical frequencies, for illustrative fibre sections lengths of 0.5 m, 1 m and 2 m. In the full 100-GHz used in the simulations, peaks as high as 9 dB above the mean were found. All spectra present randomly spaced relatively strong peaks, whose spectral width is also random and sharpens as the length is increased.

Figure 1(d) shows the distribution of peak widths (FWHM) of Rayleigh scattered light calculated in a 100-GHz spectral range. The mean width is inversely proportional to the length of the scattering fibre, and the longer the fibre section, the sharper the peak width distribution. Remarkably, the most likely spectral width reflected (e.g., 50 MHz or 0.4 pm linewidth) is that of a very weak FBG of the same length (e.g., 2 m long). A simple approach to understanding this feature is to consider the backscattering fibre section as consisting of a long fibre Bragg grating with a very small index depth and a random period over its length. The longer the grating, the sharper the reflection peaks[38].

Consider now a long lumped-gain laser cavity (e.g., 200 m) in which one of the mirrors is replaced by a short backscattering fibre

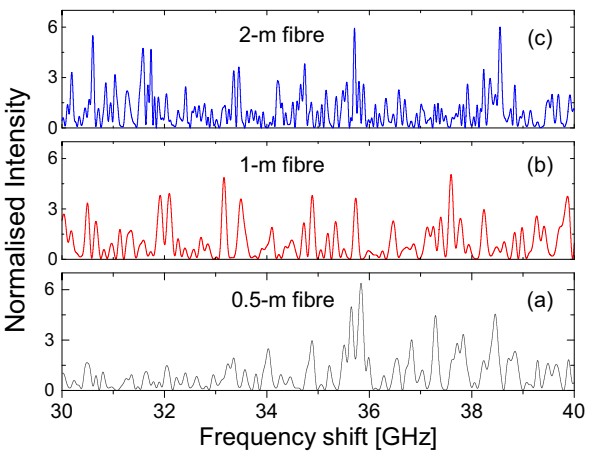
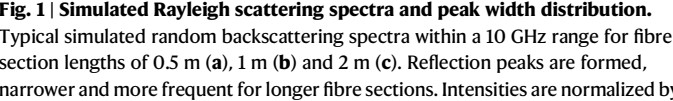
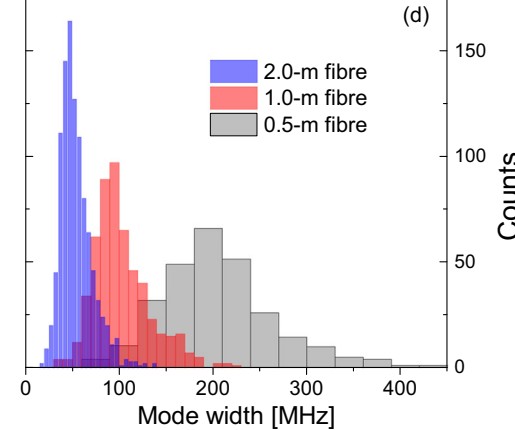

**Fig. 1 | Simulated Rayleigh scattering spectra and peak width distribution.** Typical simulated random backscattering spectra within a 10 GHz range for fibre section lengths of 0.5 m (**a**), 1 m (**b**) and 2 m (**c**). Reflection peaks are formed, narrower and more frequent for longer fibre sections. Intensities are normalized by the mean value. (**d**) Statistics of the peak widths for three lengths of backscattering fibre section calculated over a 100 GHz span. The most likely spectral width of the distributed random mirror (e.g., 50 MHz) predicted by Eq. (1) equals the width of a very weak FBG of same length (2 m).

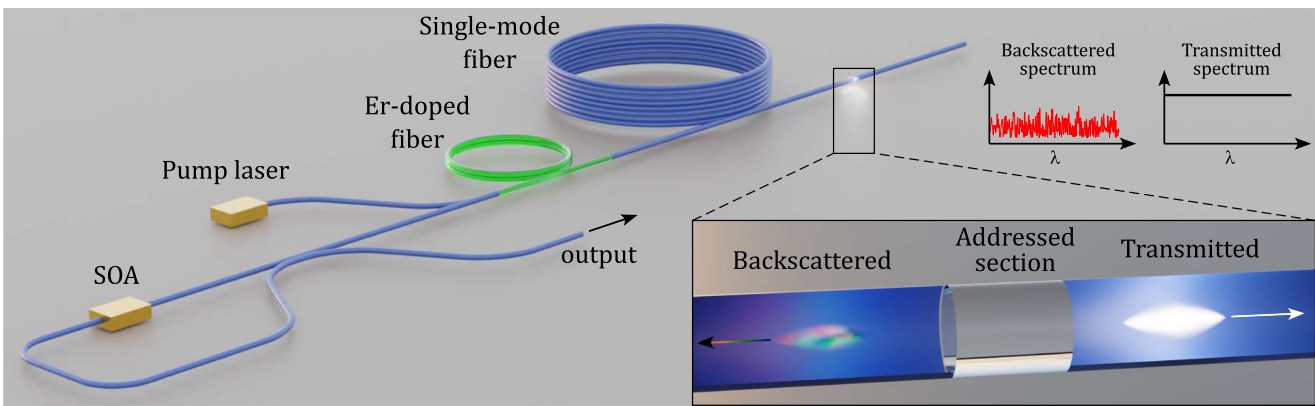

**Fig. 2 | Experiment set-up.** Schematic diagram of the mode-locked random fibre laser near but below threshold. The distributed random mirror is arbitrarily chosen as a short section of the single-mode Telecom fibre.

section (e.g., 1 m). Of course, the enormous loss of this cavity must be compensated by a huge gain, so that the laser can reach threshold. This concept is displayed in Fig. 2, where a gated semiconductor optical amplifier (SOA) generates a pulsed gain and a pumped Erbium-doped fibre (EDF) provides bidirectional CW gain. The gated-gain configuration eliminates the need to consider all backscattered light but the one from the addressed section.

Consider that a broad amplified spontaneous emission (ASE) pulse is launched by the SOA and reaches the addressed section after amplification in the EDF. The faint and randomly coloured backscattered signal coming from this section is guided back to the SOA, arriving at the very moment when another current pulse opens the gain window. When the EDF gain is sufficient, so that the spectral density at a given coherence spike of the backscattered pulse exceeds the spontaneous emission density, then this wavelength dominates the stimulated emission, depleting the gain at other wavelengths. The spectral width of the coherence spike defines the lasing linewidth. Any of these peaks accommodates hundreds of equally spaced cavity modes that could be phase-locked.

## Laser threshold

Experiments were performed with a fibre laser cavity in a sigma configuration, comprising a fibre loop and an open-ended fibre patch. The loop consisted of a gated semiconductor optical amplifier (SOA) and a length of fibre used as a delay (125 m), both connected to the two fibres of a 50/50 fused coupler, as schematically shown in Fig. 2. In some experiments when more gain was needed, an additional commercial erbium-doped fibre amplifier (EDFA) and an adjustable spectral filter were added to the loop to increase the gain and reduce the amplified spontaneous emission, respectively as detailed in Supplementary Fig. 1. The open-ended fibre section was also connected to the coupler and included a 27-m long EDFA without isolators, providing optical gain in both directions, and a passive single-mode fibre (SMF) spool that backscattered light into the loop. In most experiments, dispersion-shifted fibre was employed, but a standard Corning SMF-28 telecom fibre was also used with the inclusion of the extra amplifier that provided enough gain for laser action. The fourth leg of the coupler was used as a laser output port. The EDFA(s) provided CW gain. The SOA was gated with nanosecond current pulses from a tuneable generator and prevented light transmission while unpumped (>−60 dB). The repetition rate of the electrical pulses was adjusted in the interval 714 kHz - 1.220 MHz. As mentioned above, this allowed matching the nanosecond gain window to the arrival at the SOA of light from the chosen short section within the spool of Telecom fibre (3 cm–2 m). The length of the distributed random mirror was selected by choosing the duration of the electrical pulse applied to the SOA (e.g., 1 m for 10 ns pulses). The length of the intra-cavity delay fibre

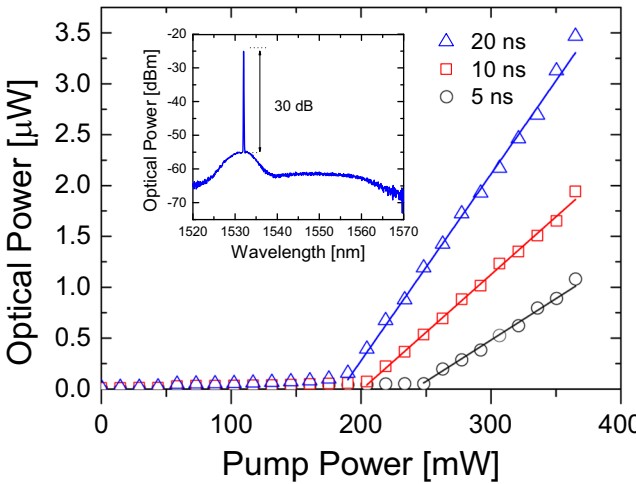

**Fig. 3 | Laser threshold.** Distinct random laser threshold for three current pulse durations. The threshold EDF pump power decreases for longer pulses, while the efficiency increases. The inset shows the optical spectrum for 20-ns pulses at full pump power.

used (125 m) ensured that doubling the shortest possible cavity length (250 m) would fall beyond any point of the piece of single-mode fibre indicated in Fig. 2. In this way, every RF frequency was univocally associated to a single cavity length and not its harmonic. The laser output was characterised in time with an amplified 26-GHz photodiode and a 3.5-GHz oscilloscope. High-resolution optical spectra were analysed with a 20-MHz frequency resolution (0.16 pm wavelength) optical spectrum analyser (OSA).

Figure 3 presents the laser average output power as a function of the EDFA pump power when the system is driven at 820 kHz with square current pulses with widths 5, 10 and 20 ns. The corresponding optical pulse durations were 3.6, 8.6 and 18.7 ns, respectively, slightly shorter than the electrical pulses, as illustrated in Supplementary Figs. 2 and 3. A distinct random laser threshold is observed in all cases[39,40] with a clear sharp laser line emerging from the broadband ASE when the threshold is reached, as shown in the inset. The higher feedback provided by longer distributed random reflectors reduces the pump power needed for lasing as more power is backscattered. Considering the sum of point losses and the cavity roundtrip gain (see Supplementary Notes), the average Rayleigh scattering capture coefficient of the fibre is insufficient to sustain laser oscillation. Indeed, a coherent reflectivity spike ~10 dB above the average is needed for the laser to reach threshold, in agreement with the model above[29]. It is worth noting that the slope of the curve increases as the

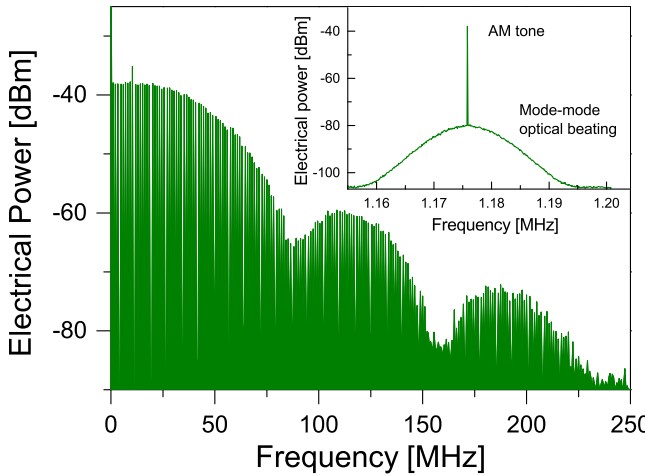

**Fig. 4 | Mode spectrum and linewidth.** Self-homodyne optical beat between the fundamental mode and the first optical sideband of the delayed replica for a ~12 ns pulse. The modulation frequency was 1.176 MHz and the number of modes within the first notch is ~86. The inset shows a zoom at around the fundamental tone, showing the optical beat and mode linewidth.

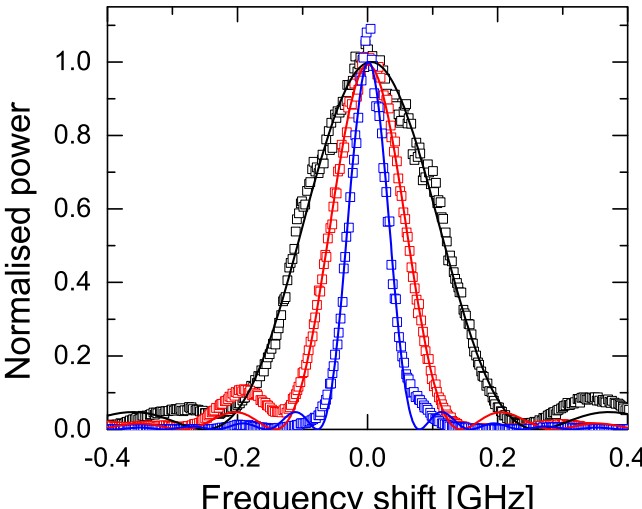

**Fig. 5 | Average optical spectra.** Laser spectral bandwidth measured with an optical spectrum analyser (50× average) for square current pulse durations 5 ns (black), 10 ns (red) and 20 ns (blue). The FWHM spectral widths fitted to $sinc^2$ functions are 227, 125 and 65 MHz, respectively, close to those expected for transform-limited pulses.

pulse duration increases, meaning that the efficiency increases with the random mirror reflectivity.

## Mode locking

Figure 4 displays the spectrum of the output signal intensity from a ~12 ns pulse, showing a series of equally spaced tones as expected for a 1.17 MHz periodic signal with top-hat shape. To demonstrate phase coherence among locked modes, a self-homodyne optical beat measurement between the pulse and a delayed replica was performed using a Mach-Zehnder interferometer (MZI) with ~100 km unbalanced paths[11]. By carefully adjusting the polarizations in each arm and the repetition rate to precisely overlap the pulse and its delayed replica, the self-homodyne spectrum could be measured. Since the spectrum is a comb of equally spaced modes, the detected self-homodyne spectrum will be given by the self-beat of all modes, centred at the baseband, plus the beat of first neighbouring modes centred at the repetition frequency tone, the beat of second neighbours at twice the repetition rate and so forth. Moreover, the detected signal will also display the frequency comb corresponding to the pulsed signal waveform, which appears even if the pulses do not overlap in time. The inset in Fig. 4 shows a zoomed measurement with high resolution (10 Hz) around the repetition rate tone. It presents a clear bell-shaped beat spectrum corresponding to the optical beat of laser modes, only detectable when pulses overlap and the polarization on both MZ arms are parallel. The same bell-shaped phase noise appears in all harmonic beat tones, including the baseband, meaning that all optical modes have the same 4.5-kHz linewidth. Interesting enough, this linewidth corresponds to the background acoustical noise in the laboratory, which affects the mode phase by sound-induced index variations in the ~200-m-long optical fibre cavity.

Considering the optical powers involved and the small length of fibres in the cavity, dispersion and nonlinear effects were estimated to be negligible in the lasing processes here. Hence, the optical field at any point inside the cavity can be written as a simple superposition of equally spaced cavity modes:

$$E(t) = \sum_k E_k \exp[j((k2\pi f + 2\pi\delta\nu_k)t + \phi_k)] \qquad (2)$$

where f is the mode spacing, $\phi_k$ is the optical phase of each mode, and $\delta\nu_k$ is the mode frequency noise giving its linewidth. Because of the

intra-cavity pulsed modulation, the time dependence of the optical field at the SOA must be written as:

$$E(t) = E_0 \sum_k M_k \exp[j((k2\pi f)t + \phi_0)] \qquad (3)$$

where $M_k$ are the Fourier coefficients and $\phi_0$ the phase of the modulation waveform. Since $\delta\nu_k \ll f$, Eqs. (2) and (3) imply that the phase $\phi_k$ of the optical modes are all locked at the SOA and match the phase $\phi_0$ of the modulation waveform.

## Spectral measurements

Figure 5 presents the experimental optical spectra of laser emission as a function of frequency shift around 1532 nm wavelength when the gated amplifier is driven with top-hat current pulses of duration 5, 10 and 20 ns. The traces displayed show an average of 50 measurements. A solid line is also displayed, indicating the best $sinc^2$ fit for the three data sets. Good agreement (R2 ~ 0.99) is observed between the experimental data and the model predictions shown in Fig. 1d. The measured bandwidths (227, 125 and 65 MHz, respectively) are approximately inversely proportional to the pulse duration, that is, to the length of the fibre section providing feedback to the random laser. Considering the case of 125 MHz bandwidth pulses at a repetition rate of 0.82 MHz, ~$10^2$ cavity modes are contained in the 3-dB optical bandwidth. The averaging used here prevents a more quantitative comparison between the measured pulses and the Fourier limit. For example, the time-bandwidth product in this case is 17% higher than the 0.89 value expected for square pulses[41].

For better synchronisation, temporal and spectral measurements were performed in single-shot acquisitions taken at the same time in the oscilloscope and spectrum analyser. Figure 6a-d shows time-domain (traces (a) and (c)) and spectral-domain (traces (b) and (d)) measurements for electrical pulses of duration 10 ns (a) and (b) and 2 ns (c) and (d). The time and spectral measurements shown were taken in single sweep mode and near synchronism. The optical pulses are shorter than the electrical ones (8.6 ns and 340 ps FWHM, respectively) and are fitted in (a) to a square pulse shape and in (c) to a Gaussian, both illustrated by a red solid line. The red solid lines in Fig. 6b and d are the corresponding Fourier transform curves, with $sinc^2$ and Gaussian shapes, respectively, with no width adjustment. When the pulse

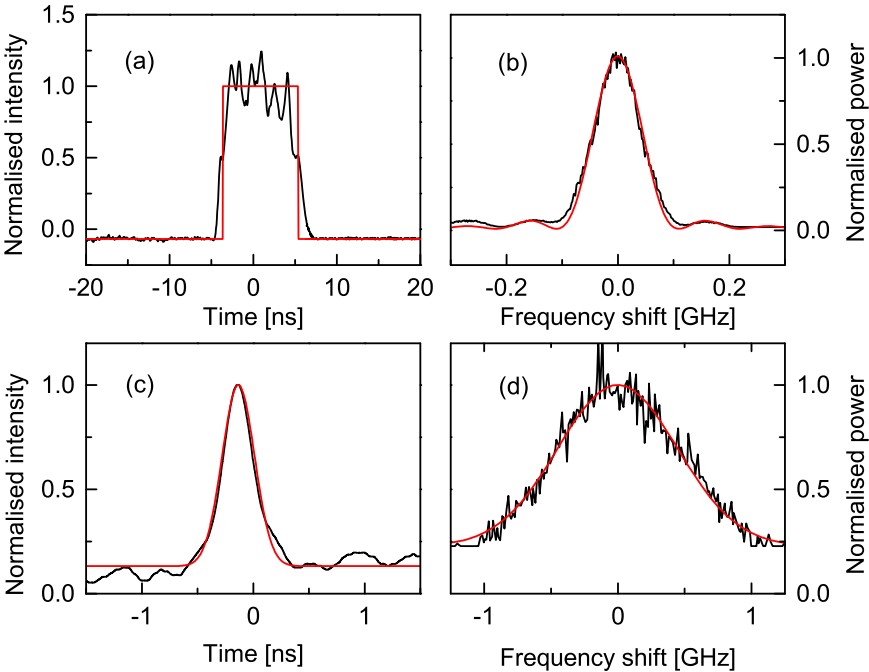

**Fig. 6 | Single shot measurements. a** Temporal and **b** spectral features of 8.6 ns FWHM laser pulse. **c** Temporal and **d** spectral characteristics of a 340 ps Gaussian pulse. Both pulses are transform-limited, as shown with the *sinc²* and Gaussian Fourier transform pairs of curves (red solid lines). All frequency shifts are measured around 1532 nm wavelength.

shapes were closer to Gaussian (e.g., Fig. 6c) than top-hat (e.g., Fig. 6a), the time-bandwidth product reduced from ~0.89 to ~0.44, as expected[41]. The good agreement of the Fourier-transform curves in red and the experimental traces in black in Fig. 6b, d demonstrate that the present mode-locked random laser can generate transform-limited pulses using the Rayleigh backscattering from a section of Telecomm fibre as short as 3.4 cm as cavity feedback (Fig. 6d). Even shorter pulses should be possible, but experiments here were electronics-limited.

More extensive investigations were conducted for various repetition rates and pulse durations. It was found that the transform limit characteristics is independent of the repetition rate, as illustrated in Supplementary Figs. 5, 7–12 and in Supplementary Table 1. While longer current pulses lead more often to pulsing at two frequencies separated by hundreds of MHz that beat in the time domain, the pulses remain transform-limited. This double-pulsing behaviour has been described in other rare-earth laser systems that lack a saturable absorber[42,43]. Single-pulse generation was more readily obtained for current pulses ≤3 ns.

## Application

Distributed temperature sensing is an important application area of optical fibres in industrial settings, to optimise production processes and minimise maintenance and repair costs. Although one Kelvin temperature sensitivity and metre-long spatial resolution suffice in many cases, such as in oil-well facilities, in other applications, such as monitoring water seepage through temperature variation in dams and dikes to prevent accidents, much better performance parameters are advantageous[44]. Typically, distributed temperature sensors (DTS) make use of the relative intensities of Stokes and anti-Stokes signals from Raman scattering in fibres, monitored via optical time domain reflectometry[45,46]. Wavelength-dependent loss due to fibre bending or ageing is a problem that single-wavelength monitoring systems do not have. Besides, the poor signal-to-noise ratio in Raman DTS is a limitation not usually encountered when measuring the output wavelength of a laser. The configurable random laser here, which allows for in situ selection of the length and bandwidth of the transform-limited probe

pulses, can be exploited in distributed temperature sensing, as an example of an application area. From Eq. (1), it is clear that the frequency spectrum shifts as the mean refractive index of the selected fibre section varies with temperature. Given the sharpness of the lasing peak, even a tiny spectral shift can be detected precisely.

As proof of principle, the open-ended telecom fibre was unwound from the spool, and a section ~1 m close to the end was immersed in water in an enclosed polystyrene box. The water temperature was initially raised, and the cooling process was monitored using a thermocouple-based thermometer and the laser wavelength. The repetition rate of the laser was tuned to 728.2 kHz so that optical feedback was provided from the section of fibre immersed in water. In order to avoid edge effects and to maximise temperature uniformity in the immersed fibre, the pulse duration was limited to 900 ps (i.e., the laser feedback came from a central 9.0-cm-long fibre section). An optical spectrum analyser was used to monitor the frequency (wavelength) shift, similar to monitoring the spectrum of simple intra-cavity FBG systems[47,48] (see Supplementary Fig. 4). Figure 7 illustrates the temperature evolution as a function of frequency shift, measured with ~4 MHz resolution (see Supplementary Methods and Supplementary Fig. 1). The frequency dependence on temperature in this small range is linear and approximately 1.19 GHz/K, similar to that found for FBG's. However, different from FBG's that only offer point sensing, distributed measurement here is accomplished by scanning the repetition rate of the laser. In addition, the mode-locked random laser also exhibits advantages over distributed sensing techniques such as Φ−OTDR, since frequency shifts are directly measured (no post-processing), have a high signal-to-noise ratio and are robust to power fluctuations. Moreover, the spectral width of the laser line is much smaller than those of typical Bragg gratings used for local temperature measurements, so that the spectral shift can be better determined (see Supplementary Methods for details).

The robustness of the experimental set-up was verified by re-starting a measurement campaign a few days later and controlling that the peak position depends only on the sample temperature (see Supplementary Fig. 6). Note that the laser cavity was protected in the

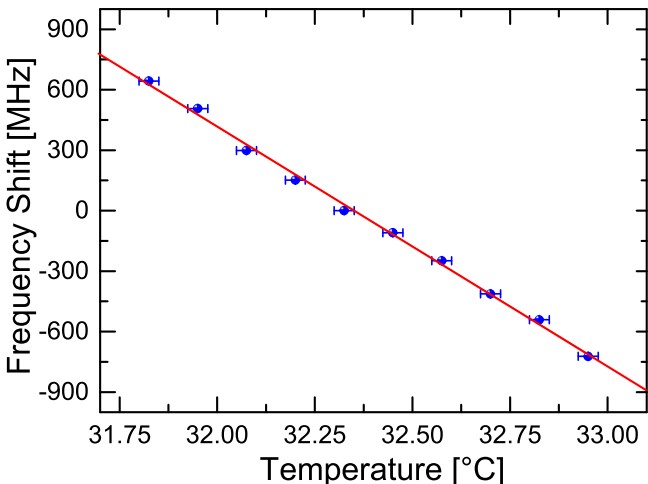

**Fig. 7 | Temperature sensing.** Rayleigh scattering in single-mode telecom fibre gives rise to transform-limited pulse generation making the laser system highly spectrally efficient. Here, 9.0 cm resolution and 0.0033 K precision are demonstrated.

laboratory but not temperature-stabilised. The spectral measurements depend almost entirely on the temperature of the particular fibre section being addressed. Other temperature-induced fluctuations affect the cavity modes but to a large extent do not alter the spectrum of the random mirror. A temperature variation of 1 K, over the full 200-m fibre circuit, would change the optical path of the cavity by a mere ~1 mm, well below the spatial resolution of the system. The spectral analysis of the backscattered light, however, can easily detect a variation two orders of magnitude smaller. It is worth noting that the relation between laser wavelength for each fibre section and the absolute temperature is unknown a priori and a calibration is needed for absolute measurements.

## Discussion

A mode-locked random fibre laser, generating transform-limited optical pulses was demonstrated. The extremely weak coloured Rayleigh backscattering of a short piece of telecom fibre is used as cavity feedback. It provides self-adjusted spectral filtering in a laser cavity, which leads to transform-limited pulse generation at various pulse durations and repetition frequencies. With such weak feedback, a gain of the order of 100 dB is required to compensate for the loss. These loss and gain values are amongst the most extreme ever reported for a laser, and this work arguably demonstrates for the first time, to the best of our knowledge, that the Rayleigh backscattering of centimetre-long telecom fibres is sufficient to achieve laser action. An estimate of the number of photons in a 340-ps pulse backscattered in the telecom fibre gives ~$10^2$ photons. It is likely that shorter pulses, well into the picosecond regime could be generated and exploited here. This means that lasing could start with an even smaller number of photons and the system could be useful to explore more fundamental questions, such as manifestations of the quantum behaviour of the laser source and how the optical pulse evolves from white light to the coherent regime in the Fourier limit. Pulses of duration under 0.3 ns were not generated for limitations in the electronics available. The nanosecond regime does, nevertheless, find important applications[49,50], as exemplified above for distributed sensing. The high-resolution measurements made possible with the transform-limited pulses here can be useful for instance in early warning of water seepage in dams. The good temperature sensitivity should allow for the use of the fibre even in a rugged cable, as necessary for field installations. One strength of the work described here is the possibility of using short sections of installed standard telecom fibre for distributed sensing. In some cases, it may be advantageous to deploy a fibre with enhanced Rayleigh scattering, as obtained for instance by exposure to UV radiation. Up to 20 dB increase in backscattering has been demonstrated[51]. This and other means[52,53] could alleviate the need for an additional amplifier in the laser cavity. In contrast to the interesting work reported in refs. 23,24, true mode-locking of a random laser is achieved here with the generation of transform-limited pulses. Unlike those publications, a single pulse per roundtrip is demonstrated in an electronically controlled cavity capable of performing distributed sensing. Furthermore, the physics of Rayleigh backscattering of coherent light is not restricted to single-mode fibres, and could also be applied to other scattering media. Given the sensitivity of the system to extremely low levels of backscattered light, its concept can conceivably be applied to distributed sensing of dilute scattering particles in free space, as a lasing-LIDAR with transform-limited resolution.

## Data availability
The data that support the findings of this study are publicly available at https://zenodo.org/records/10205014.

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

## Acknowledgements

The authors thank J. Murray and B. Redding (NRL, Washington DC) for useful discussions. Funding from CNPq (403233/2017-8, 306332/2019-1, 140701/2019-2), FAPERJ (E-26/201.200/2021), National Institute of Photonics - INCT de Fotônica (465.763/2014-6), Office of Naval Research Global (N62909-20-1-2033), K.A. Wallenberg Foundation and Swedish Research Council are gratefully acknowledged.

## Author contributions

J.W., W.M. and M.C. performed the experimental studies. P.T. performed the simulations. J.W., W.M., M.C., P.T. and A.G. contributed to the analysis.

## Funding

## Competing interests

The authors declare no competing interests.
