## [Peer Review File · Nature Communications]

REVIEWER COMMENTS

Reviewer #1 (Remarks to the Author):

In this manuscript, the authors propose and demonstrate a mode-locked random fiber laser generating transform-limited optical pulses. By means of the active modulation of the SOA (setting the repetition frequency, duration, and shape of the driving electrical pulses), it is possible to choose a short section of the single-mode telecom fiber with the wanted position and length. Weak Rayleigh backscattering from this section serves as laser feedback, and the peaks dominating the intensity spectrum of the backscattered light provide narrow spectral selectivity to the Fourier limit.

The simulation and experimental results are reasonable, but improving the experimental data is necessary before I can give a decision. My main concern is that the idea of using time gating to select a segment of fiber to provide Rayleigh backscattering for mode locking has been reported in *LASER & PHOTONICS REVIEWS*, 2018, 12(4): <https://www.webofscience.com/wos/woscc/full-record/WOS:000430301500004>. Though the authors report transform-limited optical pulse generation, there are other way of achieving such pulses, (e.g., modulating a narrow linewidth laser), what's the advantage of the method propoed by the authors.

Besides, I have a few questions about this work below.

1. Mode-locking is the basic of this work, while how to prove that mode-locking has been achieved. How many orders of longitudinal mode are locked when the pulses are generated, and what determines the pulse width, is there any pulse compressing mechanism in the proposed fiber ?
2. In Fig. 4, the authors got the transform-limited pulses at a repetition rate of 0.82 MHz. However, there are no statements about if the transform-limited pulses are generated at other repetition rates. The authors should give the necessary evidences of experimental result.
3. In a spool of single-mode telecom fiber, sections with different positions and lengths have completely different backscattering spectra / dominant peaks due to microstructural defects of the fiber during fabrication. When the repetition rate or duration of the laser is adjusted (as required for distributed sensing applications), it is difficult to determine the exact wavelength of the emission peak. Will this induce different threshold and efficiency of a laser emission. Although a bandpass filter is used, there may still be slight differences in the emission wavelengths and reflectivity of backscattering, which is very unfavorable. I would like the authors explains my concern.
4. In lines 165 to 166, I doubt the authors' explanation. The reduction in required pump power (e.g., 20 ns) may be more due to the increased duration time of SOA amplification.
5. OFDR has the advantages of high accuracy and a large dynamic range. By contrast, what is the peculiarities of the DTS method proposed in the manuscript? And what will DTS benefit from the configurable random laser proposed here?

6. Without chirp is the premise of transform-limited pulse, whether the dispersion and nonlinearity were considered/discussed in the experiment?
7. The authors should define abbreviations before using them, e.g., SOA and EDF, on page 3.
8. There's something wrong with the units of the x-coordinate in Figs. 4(a) and 4(e), the authors must verify again.

Reviewer #2 (Remarks to the Author):

The paper of Jean Pierre von der Weid et al reports the generation of fourier transform-limited pulses in random lasers, by exploiting a one dimensional fiber laser designed to have separate section for the random feedback and for the gain amplification. The random section comprises a telecom fiber providing the random interference rayleigh scattering, and simultaneously introducing consistent loss in the laser cavity. The gain section is then engineered to compensate the high loss of the random section. Mode locking is achieved with a time-gated amplification of the circulating signal synchronous to the randomly filtered feedback. I believe that this work is brilliant since a simple and elegant paradigm led to impressing results.

The authors demonstrate the reliability of their approach, as they shown how the Fourier transom pulse duration can be controlled by the length of the random feedback section over few ns to ps wide range. They also show how the temperature can be used to control the frequency emission.

The paper is well organized, well written and very detailed. I think that it can be considered for publication in nature communications after addressing the following points.

My main criticism is on the engineering of the described approach. As far as I understand, the only parameter they discuss is the length of the random section (Fig.1). I think that the paper would benefit if they were able to correlate the mode-locking with the randomness. Albeit possessing universal properties, not all the random systems are equivalent. For example, one can evaluate the amount of short-range disorder in the selected random structure, and, if present, analyze the correlation with the resulting feedback (see Biasco et al, Light Science and Applications 8, 1-13, 2019). Moreover, the author should discuss the possibility to widespread the proposed approach, i.e. synchronize the gain amplification with the random backscattering feedback to achieve transform limited mode locking, to different laser architectures.

The phrase "generally emit broadband low coherence light" in the abstract is not completely correct and should be explained better its context.

what coherence is referring? Spatial or time? The spatial coherence of random laser is indeed one key property (see for example pogna et al, Photonics Research 10 (2), 524-534, 2022). Low spatial coherence

has been reported in engineered random lasers, making random lasing emission very appealing for speckle free imaging application.

On the other hand, mode locking stems from time coherence, and as its pointed out in the paper and in other articles duly cited in the paper, it is indeed possible in random lasers (see for example A.DiGaspare et al *Advanced Science* 9 (28), 2200410). The authors themselves provide as a possible application the LIDAR with transform-limited resolution.

REVIEWER COMMENTS

Reviewer #1:

In this manuscript, the authors propose and demonstrate a mode-locked random fibre laser generating transform-limited optical pulses. By means of the active modulation of the SOA (setting the repetition frequency, duration, and shape of the driving electrical pulses), it is possible to choose a short section of the single-mode telecom fibre with the wanted position and length. Weak Rayleigh backscattering from this section serves as laser feedback, and the peaks dominating the intensity spectrum of the backscattered light provide narrow spectral selectivity to the Fourier limit. The simulation and experimental results are reasonable, but improving the experimental data is necessary before I can give a decision. My main concern is that the idea of using time gating to select a segment of fibre to provide Rayleigh backscattering for mode locking has been reported in LASER & PHOTONICS REVIEWS, 2018, 12(4): <https://www.webofscience.com/wos/woscc/full-record/WOS:000430301500004>.

We thank the referee for pointing out to us this interesting paper [1a]. Indeed, the material in it is highly relevant to our work. We now included it in our reference list. Significant differences exist between Weiwei Pan's paper and ours. There, the gain medium is Raman at 1 μm , in ours a SOA and EDFA at 1.5 μm . There, the laser produces picosecond pulses, in ours, typically nanoseconds. There, the spectrum is several nanometers wide, in ours four orders of magnitude narrower. In our work, we generate transform-limited pulses, in theirs not. Besides, the regime of operation differs, as explained below.

In their work, Pan et al report the stable generation of picosecond pulses from a random feedback laser. The stability of their ingenious system rests on the fact that a pump pulse will always meet a "pulse continuum" and amplify. With their pump frequency 28 MHz, i.e., one pulse at every 35 ns and a minimum cavity roundtrip time 300 ns (in the 60 m Raman fibre alone), they operate their laser with several pulses circulating in the cavity at once. This is valuable to increase the average power and the effective gain. Any picosecond laser pulse generated in their laser scatters at many 4.5 mm long sections of the SMF28 fibre and meet a cascade of pump pulses delayed from each other by 35 ns. Reciprocally, under every pump pulse there are contributions from many SMF sections (adding to 25.6 m after 20 km). The roundtrip time of any pulse is not easily defined, because the pulse integrates the contribution from many scattering elements separated in space and time.

In our work, in contrast, we have a single pulse circulating in the laser cavity. The laser pulse is repetitively amplified at a single roundtrip time associated with that unique pulse. This means that the pulse is repetitively reflected by a single section of Rayleigh backscattering fibre, with its unique Rayleigh fingerprint [2a]. In our work, we have a single backreflector for every pulse that we generate. The length of the backreflector is chosen by the electrical pump pulse duration and its position chosen by the repetition frequency. This allows us to have a repetitive spectral mirror and to generate transform limited pulses. The repetitive amplification at a constant roundtrip and the repetitive reflection at a mirror with a fixed Rayleigh fingerprint allows our pulses to synchronously mode-lock in the classical way.

We are grateful to the referee for having pointed out this important work [1a]. We now refer to it in our introduction, with a sentence presenting the article for the interested readers.

Though the authors report transform-limited optical pulse generation, there are other way of achieving such pulses, (e.g., modulating a narrow linewidth laser), what's the advantage of the method proposed by the authors.

We agree with the referee. Indeed, transform limited pulses can be obtained by using an external modulator and a narrow linewidth laser, the spectral purity being limited by the chirp of the modulator.

One advantage of our laser as compared with the external modulator approach is that all the gain available in our laser is used for pulse generation (and the output signal). In contrast, an external modulator cuts out all energy that is not within the transform limited pulse carved.

Another clear advantage of our approach is that our laser embodies an intracavity distributed fibre sensor, as shown in our paper. The Rayleigh scattering fibre is an integral part of the cavity, and the wavelength of the transform-limited pulse formed can be used for distributed temperature sensing (DTS) with excellent resolution. When the wavelength is determined by a modulated narrow linewidth laser, it is fixed from the start and therefore not easily used for DTS.

Our laser is obviously not the single source of transform-limited pulses, but it clearly adds to the useful sources available.

Besides, I have a few questions about this work below.

1. Mode-locking is the basic of this work, while how to prove that mode-locking has been achieved. How many orders of longitudinal mode are locked when the pulses are generated, and what determines the pulse width, is there any pulse compressing mechanism in the proposed fibre ?

We thank the referee for this important consideration, which made us discuss the fundamental aspects of mode-locking in greater detail. We introduced a new measurement and a new mathematical argumentation to support the statements about mode-locking in a new subsection. All modes of the cavity are locked. Depending on modulation frequency and pulse duration some tens to a few hundreds of modes have the same phase. We illustrate the modal distribution and the phase noise of the modes in the new Fig. 5 seen here, with the inset showing the small phase noise of all modes, ~5 kHz. The self-homodyne measurement is explained in detail in the text of the paper.

We see that for 12 ns pulses (at 1.176 MHz), there are 86 modes within the spectrum measured at the first notch of the distribution (87 MHz). The number of lasing modes increases for shorter duration pulses and lower repetition frequencies.

The optical field at any point inside the cavity can be written as a superposition of equally spaced cavity modes:

$$E(t) = \sum_k E_k e^{j(k2\pi f + 2\pi\delta\nu_k)t + \phi_k} \quad (2)$$

where f is the mode spacing, ϕ_k is the optical phase of each mode, and $\delta\nu_k$ is the mode frequency noise giving its linewidth. Because of the intra-cavity pulsed modulation, the time dependence of the optical field at the SOA must be written as:

$$E(t) = E_0 \sum_k M_k e^{j(k2\pi f)t + \phi_0} \quad (3)$$

where M_k are the Fourier coefficients and ϕ_0 the phase of the modulation waveform. Since $\delta\nu_k \ll f$, Eq. (2) and (3) imply that the phase ϕ_k of the optical modes are all locked at the SOA and match the phase ϕ_0 of the modulation waveform.

We believe and hope that these considerations, now introduced in the main body of our manuscript, explain more clearly the number of modes locked and the main mechanism in operation, which is the periodic gating of the SOA.

2. In Fig. 4, the authors got the transform-limited pulses at a repetition rate of 0.82 MHz. However, there are no statements about if the transform-limited pulses are generated at other repetition rates. The authors should give the necessary evidences of experimental result.

We thank the referee for asking this question. Transform limited pulses are generated along the full repetition range covering the length of the measured fibre, from ~ 714 kHz ~ 1.22 MHz. The duration of the electrical current pulse largely determines the duration of the optical pulse, the inverse of which gives the spectral laser bandwidth within a factor in the range 0.31 to 0.89 associated to the shape of the pulse (sech², Gaussian, square etc). We added two sentences to the text to clarify this point:

“The length of the intracavity delay fiber used (125 m) ensured that doubling the shortest possible cavity length (250 m) would fall beyond any point of the piece of single mode fiber indicated in Fig. 3. In this way, every RF frequency was univocally associated to a single cavity length and not its harmonic.”

“...More extensive investigations were conducted for various repetition rates and pulse durations. It was found that the transform limit characteristics is independent of the repetition rate.”

3. In a spool of single-mode telecom fibre, sections with different positions and lengths have completely different backscattering spectra / dominant peaks due to microstructural defects of the fibre during fabrication. When the repetition rate or duration of the laser is adjusted (as required for distributed sensing applications), it is difficult to determine the exact wavelength of the emission peak. Will this induce different threshold and efficiency of a laser emission. Although a bandpass filter is used, there may still be slight differences in the emission wavelengths and reflectivity of backscattering, which is very unfavorable. I would like the authors explains my concern.

The referee’s statement is correct. The precise optical frequency generated by a given section is unpredictable a priori, and depends on the very properties of the addressed section, the Rayleigh fingerprint, which is fixed for that section. Different fibre sections lase at different wavelengths within the

bandpass filter that is much broader than the laser emission and simply restricts the gain to a limited spectral region, avoiding multimode operation. Note that the filter width is typically ≤ 1 nm, so the laser wavelength is still relatively well defined. Rather than exhibiting widely varying thresholds, the laser chooses the spectral peak that gives highest reflectivity within the filter width, and lasing occurs at that wavelength.

This laser source finds applications in distributed sensing rather than as a stable narrow linewidth source, which seems to be the focus of the referee's concern. Similar to a fibre laser incorporating an array of FBGs as backreflectors [3a-5a], the lasing wavelength is defined by the environmental conditions of the addressed section. The method requires calibration since it measures variations of the fibre properties, not their actual value. Hence, when the fibre is installed in its initial state, a wavelength emission function is generated, mapping the laser wavelength for each fibre section (defined by the repetition rate). At a later time, a new function can be measured and the difference between optical emission wavelengths corresponding to each fibre section provides the variation of the distributed values of the parameter of interest, e.g. the fibre temperature.

We added a sentence to the text clarifying this point:

“It is worth noting that the relation between laser wavelength for each fibre section and the absolute temperature is unknown a priori and a calibration is needed for absolute measurements.”

Summing up, the laser described does not excel by its wavelength stability, but rather, by its potential use in sensing and as a source of transform limited pulses. If the laser is to have a constant wavelength within ~ 1 pm, then the packaging of the laser and the fibre need to be temperature stabilized to a small fraction of a degree.

4. In lines 165 to 166, I doubt the authors' explanation. The reduction in required pump power (e.g., 20 ns) may be more due to the increased duration time of SOA amplification.

Like the referee, we also believe that the reduction in required pump power is due to the increased duration time of the SOA pulse. Longer times result in more average power because the duty cycle is increased. This leads to increasing efficiencies and smaller threshold powers, as shown in Fig. 3. The way we had phrased lines 165 and 166 led to misunderstanding. We rewrote those lines now.

5. OFDR has the advantages of high accuracy and a large dynamic range. By contrast, what are the peculiarities of the DTS method proposed in the manuscript? And what will DTS benefit from the configurable random laser proposed here?

We agree with the referee that meter-length or even higher resolutions in OFDR can be obtained ensuring high linearity of the frequency sweep or accurate nonlinearity compensation methods such as auxiliary interferometer data acquisition trigger. The combination of high resolution with long distances requires either ultra-low phase noise sources to avoid sinking the Rayleigh scattering signal in the noise floor or the use of fibre with high Rayleigh scattering [6a,7a].

A simpler wavelength-based approach to measure temperature is to use a spectrometer in conjunction with a laser with (intracavity) fibre Bragg gratings, or as in our manuscript, the Rayleigh fingerprint of a section of fibre. With an FBG array, multipoint sensing is obtained, where temperature readings are possible at every FBG [5a]. Here, the laser chooses the highest spectral peak in the addressed fibre section and locks to it. This becomes a true distributed sensor method because every section (as

short as a few centimetres) lases at its spectral peak, without a FBG having to be written in it. The method is simple and highly precise, as shown in the DTS measurements of our manuscript.

6. Without chirp is the premise of transform-limited pulse, whether the dispersion and nonlinearity were considered/discussed in the experiment?

We thank the referee for this important observation. Indeed, nonlinearity and dispersion could affect the pulses generated and an estimate should be made of the severity of these two effects. We use below the estimates from Agrawal's book on nonlinear fibre optics [8a].

Nonlinearity: The average power of the laser was typically 3.5 μ W at 1 MHz at the output port of the laser and 0.7 mW at the through port after the sensing fibre. The peak power for 10 ns pulses was 100x higher, i.e., typically 0.07 W. Considering that a phase shift of π radians is induced at a length $L_{nl} = 1/\gamma P_o$, where P_o is the power of the pulse and $\gamma = 2 \text{ W}^{-1}/\text{km}$ the nonlinear coefficient of SMF fibre, the nonlinear length for our pulse is ~ 7 km. This is two orders of magnitude longer than the fibre lengths used, so we do not expect the presence of significant nonlinear effects.

Dispersion: Likewise, the dispersion length for which dispersion starts to play a role is estimated from $L_D = T_o^2/|\beta_2|$, where the absolute value of the group velocity dispersion parameter is typically 20 ps^2/km . The shortest pulses used were 250 ps long. The dispersion length in this case exceeds 3000 km, so dispersion does not play a measurable role in the physics of the laser discussed here. \sim

In the new subsection concerning mode locking we added a sentence clarifying the assumptions involved:

...Considering the optical powers involved and the small length of fibres in the cavity, dispersion and nonlinear effects were estimated to be negligible in the lasing processes here. Hence, the optical field inside the cavity can be written as a simple superposition of equally spaced cavity modes:

7. The authors should define abbreviations before using them, e.g., SOA and EDF, on page 3.

We thank the referee for the observation and corrected the text accordingly.

8. There's something wrong with the units of the x-coordinate in Figs. 4(a) and 4(e), the authors must verify again.

The referee is right, this was an error that is now corrected. All x-coordinates are given in GHz. Thank you.

References [reviewer a]

[1a] "Ultrafast Raman fibre Laser with Random Distributed Feedback", Weiwei Pan, Lei Zhang, Huawei Jiang, Xuezhong Yang, Shuzhen Cui, Yan Feng, W. Pan, X. Yang, Laser Photonics Rev. 2018, 12, 1700326. DOI: 10.1002/lpor.201700326

[2a] "Rayleigh-Based Distributed Optical Fibre Sensing", L. Palmieri, L. Schenato, M. Santagiustina and A. Galtarossa, Sensors 2022, 22, 6811. <https://doi.org/10.3390/s22186811>

[3a] "Multiplexed fibre Bragg grating fibre-laser strain-sensor system with mode-locked interrogation," A. D. Kersey and W. W. Morey, Electron. Lett. 29(1), 112–114 (1993).

[4a] "Time- and Wavelength-Division Multiplexing of FBG Sensors Using a Semiconductor Optical Amplifier in Ring Cavity Configuration," W. H. Chung, H.-Y. Tam, P. K. A. Wai, and A. Khandelwal, IEEE Photon. Technol. Lett. 17(12), 2709–2711 (2005).

[5a] "Intracavity interrogation of an array of fiber Bragg gratings", W. Margulis, R. Lindberg, F. Laurell and G Hedin, Opt. Express Vol. 29, No. 1 (2021).

[6a] "Rayleigh scatter based order of magnitude increase in distributed temperature and strain sensing by simple UV exposure of optical fibre". Loranger, S., Gagné, M., Lambin-Iezzi, V. et al., Sci Rep 5, 11177 (2015). <https://doi.org/10.1038/srep11177>

[7a] "Improving OFDR distributed fibre sensing by fibres with enhanced Rayleigh backscattering and image processing," Q. Wang et al., in IEEE Sensors Journal, vol. 22, no. 19, pp. 18471-18478, 1 Oct.1, 2022, doi: 10.1109/JSEN.2022.3197730.

[8a] "Nonlinear fiber optics", G. Agrawal 4th ed, Academic Press, Elsevier, ch 3 and 4 (2007).

Reviewer #2:

The paper of Jean Pierre von der Weid et al reports the generation of fourier transform-limited pulses in random lasers, by exploiting a one-dimensional fibre laser designed to have separate section for the random feedback and for the gain amplification. The random section comprises a telecom fibre providing the random interference rayleigh scattering, and simultaneously introducing consistent loss in the laser cavity. The gain section is then engineered to compensate the high loss of the random section. Mode locking is achieved with a time-gated amplification of the circulating signal synchronous to the randomly filtered feedback. I believe that this work is brilliant since a simple and elegant paradigm led to impressing results.

The authors demonstrate the reliability of their approach, as they shown how the Fourier transom pulse duration can be controlled by the length of the random feedback section over few ns to ps wide range. They also show how the temperature can be used to control the frequency emission. The paper is well organized, well written and very detailed. I think that it can be considered for publication in nature communications after addressing the following points.

My main criticism is on the engineering of the described approach. As far as I understand, the only parameter they discuss is the length of the random section (Fig.1). I think that the paper would benefit if they were able to correlate the mode-locking with the randomness. Albeit possessing universal properties, not all the random systems are equivalent. For example, one can evaluate the amount of short-range disorder in the selected random structure, and, if present, analyze the correlation with the resulting feedback (see Biasco et al, Light Science and Applications 8, 1-13, 2019). Moreover, the author should discuss the possibility to widespread the proposed approach, i.e. synchronize the gain amplification with the random backscattering feedback to achieve transform limited mode locking, to different laser architectures.

We thank the referee for this interesting comment. Indeed, random systems vary widely in character, and in some lasers, it is possible to artificially alter the degree of randomness of the scattering centres and the short-range disorder. This is the case when the scattering comes from nanostructures fabricated on a semiconductor surface, as in the example given by the referee [1b]. Post-processing of the fibres used in our work may lead to a change in the short-range order, e.g. by heat or by UV exposure [2b, 3b]. The work we describe in our paper makes use of the scattering in single (transverse) mode optical fibres. We focus our work on the use of commercial fibres that could be easily implemented in a laser cavity and used for sensing by the readers of our manuscript.

It should be stressed, however, that although the laser cavity proposed uses a random reflector (backscattered light from the addressed section), it relies on a deterministic path defined by the roundtrip over the cavity components. Mode locking is not correlated to randomness here, it is only possible in our case because of the deterministic characteristic of the cavity modes. If the cavity modes were random there would be no equally spaced optical modes to be locked. The 10^2 mode-locked modes in the spectrum fit under the spectral width of the reflectivity spikes associated to the length of scattering fibre (defined by the pulse duration). A new Figure 5 has been added to the manuscript illustrating the tens-hundreds of modes of the cavity, making this point clearer.

As for the valuable suggestion of the referee, for us to propose other laser architectures to achieve transform limited pulses and widen the proposed approach, we refrained from speculating in our manuscript because the available gain media that we explored here (Erbium-doped amplifiers and semiconductor optical amplifiers that can be gated withing nanoseconds) are quite unique components developed for telecommunications and that exhibit excellent properties and easy access. Although we may come up with other laser systems, we believe that this arrangement is by far the simplest and most useful that can be implemented.

We added the reference suggested by the referee to the revised text, in the context of spatial mode locking and thank the referee for the comment.

The phrase “generally emit broadband low coherence light” in the abstract is not completely correct and should be explained better its context.

what coherence is referring? Spatial or time? The spatial coherence of random laser is indeed one key property (see for example pogna et al, *Photonics Research* 10 (2), 524-534, 2022). Low spatial coherence has been reported in engineered random lasers, making random lasing emission very appealing for speckle free imaging application.

On the other hand, mode locking stems from time coherence, and as its pointed out in the paper and in other articles duly cited in the paper, it is indeed possible in random lasers (see for example A.DiGaspare et al *Advanced Science* 9 (28), 2200410). The authors themselves provide as a possible application the LIDAR with transform-limited resolution.

We thank the referee for the comment that the concept of coherence was not well defined in the abstract of our manuscript. We removed the sentence in the abstract so that it could be shorter and clearer. Besides, we now add the two references [4b, 5b] on locking the spatial modes in the introduction, where it is possible to add the publications mentioned by the referee.

References [Reviewer b]

[1b] Biasco, S., Beere, H.E., Ritchie, D.A. *et al.* Frequency-tunable continuous-wave random lasers at terahertz frequencies. *Light Sci Appl* 8, 43 (2019). <https://doi.org/10.1038/s41377-019-0152-z>

[2b] “Rayleigh scatter based order of magnitude increase in distributed temperature and strain sensing by simple UV exposure of optical fibre”. Loranger, S., Gagné, M., Lambin-lezzi, V. et al., *Sci Rep* 5, 11177 (2015). <https://doi.org/10.1038/srep11177>

[3b] “Improving OFDR Distributed Fibre Sensing by Fibres with Enhanced Rayleigh Backscattering and Image Processing,” Q. Wang et al., in *IEEE Sensors Journal*, vol. 22, no. 19, pp. 18471-18478, 1 Oct.1, 2022, doi: 10.1109/JSEN.2022.3197730.

[4b] Pogna, E. A. A., di Gaspare, A., Reichel, K., et al. Spatial coherence of electrically pumped random terahertz lasers. *Photonics Research* 10, 524-534 (2022). <https://doi.org/10.1364/PRJ.440463>

[5b] Di Gaspare, A., Pistore, V., Riccardi, E., et al. Self-induced mode-locking in electrically pumped far-infrared random lasers. *Advanced Science* 10, 2206824 (2023).

Additional alterations, not requested by the referees [c]:

Besides changes implemented in response to the referees' comments, we introduced some corrections to the text illustrated in blue, to improve the English text and clarify some confusing sentences, in particular in the Introduction.

We also added one reference to early work by Letokov, Ambartumyan and co-workers [1c] on an experimental demonstration of random lasing from 1996 that we were unaware of when we submitted the paper. Likewise, we added a reference to work from University of Padova [2c], where the concept of "Rayleigh fingerprint" is introduced. This concept is used in our paper, and now a reference is given.

We also took the liberty of separating Fig. 1a and 1b into the new Figs 1 and 2. Likewise, Fig. 4a now becomes a separate Fig. 6. We feel that this makes it possible to have dedicated captions that explain the results of the paper in a clearer way, so that our manuscript is improved.

Finally, we changed the graphics and the formatting of the text and Fig. captions to comply with the requirements of Nature Communications.

We hope that the Editorial Board now finds the paper acceptable in its present format. We send a version with all changes marked and one with all changes accepted.

References [c]

[1c] Ambartsumyan, R. V., Basov, N. G., Kryukov, P. G., et al. A laser with a nonresonant feedback. *IEEE J. Quantum Electronics*, QE-2, 442-446 (1966)

[2c] Palmieri, L.; Schenato, L.; Santagiustina, M.; Galtarossa, A. Rayleigh-Based Distributed Optical Fiber Sensing. *Sensors* 22, 6811 (2022). <http://doi.org/10.3390/s22186811>

REVIEWER COMMENTS

Reviewer #1 (Remarks to the Author):

In the revised manuscript and response letter, the authors have addressed some of my concerns. However, to strengthen the message of the work, I still have some queries, and the authors should focus on the following issues.

1. As for the second question I mentioned before, the authors show that “Transform limited pulses are generated along the full repetition range covering the length of the measured fibre, from ~ 714 kHz – ~ 1.22 MHz” and “More extensive investigations were conducted for various repetition rates and pulse durations. It was found that the transform limit characteristics is independent of the repetition rate”. However, there is still no necessary pieces of evidence of experimental results or other data at other repetition rates.

2. In lines 140 and 141, the authors say, “The corresponding optical pulse durations were 3.6, 8.6, and 18.7 ns, respectively”. However, the time domain figures of those pulses are absent, and the author should provide them.

3. As for the sixth question I mentioned, the authors show that “The dispersion length in this case exceeds 3000 km”. 3000 km is much longer than the fiber used in the experiment, however, the pulse circulates periodically in the cavity so can the dispersion be ignored? Besides, if the authors observe transform-limited pulse, what’s the role of dispersion and nonlinear effect for the mode locked pulses at different point of reflection? I think these points are import and need to be further discussed both experimentally and theoretically.

4. In response to my previous question #3, the authors claimed that the Rayleigh scattering spectrum is fixed for a given fiber section. However, I am concerned that the spectrum could be affected by, for example, slight mechanical perturbations, causing the central wavelength of the lasing spectrum to fluctuate within the spectral region limited by the bandpass filter. I think the authors should demonstrate the long-term stability of this laser in both the temporal and spectral domains.

5. In simulating the Rayleigh scattering spectra of optical fiber sections with different lengths, how is “ $d = 5$ mm” chosen so as to correspond to the experiment?

6. The novelty and significance of the proposed should be make more clear, especially, comparing with former works (6. The novelty and significance of the proposed should be make more clear, especially, comparing with former works,

e.g., Refs [1] “Ultrafast Raman fiber Laser with Random Distributed Feedback,” *Laser Photon. Rev.*, 12 (2018). [2] “Quasi mode-locking of coherent feedback random fiber laser,” *Sci Rep*, 6, 39703 (2016), and [3] “Characterization of a FBG sensor interrogation system based on a mode-locked laser scheme,” *Opt. Express* 25, 24650-24657 (2017).

Reviewer #2 (Remarks to the Author):

I believe that the authors have addressed the issues raised by the two referees in a satisfactory way.

In particular, they clarified to me that the random scattering for the mode generation and the mode locking are two separate mechanisms. Although there are several reports demonstrating how mode locking in random lasers may rely entirely on randomness (see refs 13 and 14 of their manuscript), in their rebuttal they explained how in this case Mode locking is not correlated to randomness, but it is only possible to the coherent feedback provided by the round-trip cavity path. This is a fundamental distinction, that doesn't lead to classify their approach as "not random", but help separate their proof of concept laser architecture from the other.

I think that the paper is significantly improved and its now acceptable for publication in nature communications.

REVIEWER COMMENTS and AUTHORS REPLIES

Reviewer #1 (Remarks to the Author):

In the revised manuscript and response letter, the authors have addressed some of my concerns. However, to strengthen the message of the work, I still have some queries, and the authors should focus on the following issues.

1. As for the second question I mentioned before, the authors show that “Transform limited pulses are generated along the full repetition range covering the length of the measured fibre, from ~ 714 kHz – ~ 1.22 MHz” and “More extensive investigations were conducted for various repetition rates and pulse durations. It was found that the transform limit characteristics is independent of the repetition rate”. However, there is still no necessary pieces of evidence of experimental results or other data at other repetition rates.

We thank the referee for following up on this point. Indeed, the claim made was that transform limited pulses were generated independent of the repetition rate chosen. This had been observed by us but not documented. We went back to the laboratory and newly recorded the spectral and temporal characteristics of laser pulses similar to each other generated at various repetition rates, to be able to compare their time-bandwidth products. The laser pulses were measured with the instrumentation used previously. We attach at the end of this document the details of six pairs of spectra/time domain measurements, over the range of interest. We now incorporate into the Supplementary Material the summary of our findings with the graph and table below, as well as the raw measurements with the

fittings, so that an interested reader can access all the data.

The time-bandwidth product was calculated and compared for pulse repetition frequencies 0.7, 0.8, 0.9, 1.0, 1.1 and 1.2 MHz, covering the range studied. The pulse duration chosen was approximately 2 ns, and the pulse profile was approximately square. Consistently, the power spectra of the pulses (which was collected in a single sweep without averaging) was fitted to a function Sinc^2 . Limitations in the available electronics caused the current pulses to have some ringing and not have a perfectly flat profile. The average time-bandwidth product measured was 0.83 instead of the 0.89 expected in the ideal case of perfect Sinc^2 spectra and perfectly square pulses. The $\sim 7\%$ lower product than for the ideal case can be assigned to the finite rise- and fall-time of the pulses.

Table with the summary of measurements

Frequency (kHz)	Duration FWHM	Spectral width FWHM	Product
1203	2.10	3.12	0.84
1105	2.22	2.91	0.83
1003	2.00	2.64	0.68
903	1.73	4.00	0.89
805	1.83	3.50	0.82
735	1.85	3.90	0.92
Average = 0.83; Standard deviation $\sigma = 0.083$;			

The plot below shows the experimental values of the time-bandwidth product measured at different laser repetition frequencies. The error bar is the standard deviation calculated for all measurements in this graph. The plot lacks a clear trend, which confirms that within the experimental error, the time-bandwidth product is independent of the repetition rate of the laser.

Time-bandwidth product of laser pulses generated at various repetition frequencies.

2. In lines 140 and 141, the authors say, “The corresponding optical pulse durations were 3.6, 8.6, and 18.7 ns, respectively”. However, the time domain figures of those pulses are absent, and the author should provide them.

We thank the referee for this careful review of our paper, making sure we do not make unsubstantiated claims. We had already included time-domain traces in Figure 2 of the Supplementary Information, sent when the paper was first submitted. They confirm that the optical pulse generated is slightly shorter than the current pulse applied to the SOA, as stated. We should have referred to those results in the supplementary material in the text, but do so only now, on lines 140 and 141.

We show below the time domain traces of the measurements associated with the spectra of Figure 6 of the main text, as requested by the referee. The Figure on the left shows voltage pulses into the 50 Ω oscilloscope that are a replica of the current pulses applied to the SOA. The noise seen at 15 ns is due to a spurious reflection and does not appear on the optical pulses. On the right, we display the

optical pulses measured for current pulses of 5 ns, 10 ns and 20 ns. Their width is 3.6, 8.6 and 18.7 ns, respectively.

Left: replica of electrical pulses applied to SOA of 5, 10 and 20 ns; Right: Optical pulses generated, with durations 3.6 ns, 8.6 ns and 18.7 ns, respectively.

3. As for the sixth question I mentioned, the authors show that “The dispersion length in this case exceeds 3000 km”. 3000 km is much longer than the fibre used in the experiment, however, the pulse circulates periodically in the cavity so can the dispersion be ignored? Besides, if the authors observe transform-limited pulse, what’s the role of dispersion and nonlinear effect for the mode locked pulses at different point of reflection? I think these points are import and need to be further discussed both experimentally and theoretically.

We once again thank the referee for the comments on the nonlinearity and dispersion of the fibre in the cavity. We agree that these effects could indeed affect the pulse duration and the spectrum, as would be expected in the case of very short (ps or fs) pulses of high intensity. The referee is careful in noticing that the pulse recirculates in the cavity: the cavity length of the laser is only 0.3 km (and as for the different points of reflection, adjustable from 164 m to 280 m). Although this is short and the laser cavity is extremely lossy because one mirror reflects -80 dB or less, given enough gain a pulse could in principle circulate 10000 times before exiting. The effective propagation length could in principle approach 3000 km, when dispersion becomes important.

Here, however, the photon lifetime is 200 μ s, determined from the mode bandwidth (measured to be $\Delta\nu \approx 5$ kHz in the inset of Fig. 5 in the revised manuscript). This corresponds to a propagation length of 40 km. This is still two orders of magnitude less than needed for dispersion to become apparent.

More important, however, is the robust mechanism for the generation of transform-limited pulses. The pulse duration is given by the current pulse applied to the SOA (e.g., 10 ns), which in turn defines the width of the random spectral peaks of the distributed mirror (e.g., ~ 112 MHz). At every reflection, this mirror works as a strong narrow spectral filter, and the SOA as a strong time-gate. Cumulative effects are thus hindered by the filtering experienced at every roundtrip, both spectrally and temporally. The optical pulses generated cannot widen temporally in successive roundtrips because of the gating provided by the SOA, which cuts any optical emission outside the nanosecond current pulse. Likewise, it cannot widen spectrally accumulating nonlinear phase modulation because of the spectral filter that cleans the spectrum at every reflection.

Both these laser filtering characteristics can effectively limit any eventual broadening caused by dispersion or nonlinearity. Consistently, the pulses generated are measured to be transform limited.

As regards to the different points of reflection, we remark that our original laser design had two outputs, one just after the SOA through the 50% fibre coupler, and one at the end of the sensing fibre, after one pass in one of the Erbium doped amplifiers. We never observed a difference in pulse duration between these outputs, and the spectra measured differed only in their intensity (and the level of continuous wave ASE under the pulses because of the EDFA). This is consistent with the fact that the pulse duration and spectral content does not vary significantly along the cavity.

Since our experiments show transform-limited pulses, two conceivable regimes can be in operation: 1) A soliton regime, when dispersion and nonlinearity are present but cancel each other exactly and 2) A regime when both dispersion and nonlinearity are negligible. Everything points to the latter rather than the former regime. It is difficult to see how a square pulse (and not a sech^2) could satisfy a soliton condition in an optical fibre. The edges are much more prone to nonlinear effects than the centre of the pulse, leading to chirp and a deviation from the transform-limited condition. The laser does not generate solitons, also because of the exceedingly long L_D , well above the photon lifetime in the cavity. The remaining alternative is that in the laser described and characterized here both dispersion and nonlinearity are negligible, as the calculation of L_D and L_{NL} indicate. Once again, we attribute this to the nanosecond or sub-nanosecond pulse durations and picometres bandwidths involved, a regime that differs significantly from the ps and fs regimes, where soliton effects ought to be present.

Thanking again the referee for the comment, we add a short section to the Supplementary Information of our manuscript to stress the conceivable effects of dispersion and nonlinearity and the lack of such effects for the nanosecond pulses of our laser.

4. In response to my previous question #3, the authors claimed that the Rayleigh scattering spectrum is fixed for a given fibre section. However, I am concerned that the spectrum could be affected by, for example, slight mechanical perturbations, causing the central wavelength of the lasing spectrum to fluctuate within the spectral region limited by the bandpass filter. I think the authors should demonstrate the long-term stability of this laser in both the temporal and spectral domains.

The referee is right that this is a matter of concern. Just as in a laser with a 1-m long FBG (equivalent to the laser here with 10 ns pulses) the spectral peak depends on temperature and on bending the fibre or strain on it. As the referee anticipates, even slight thermal or mechanical fluctuations will make the wavelength drift, as expected in a sensitive sensor system. The claim we make is that the laser allows for sensitive measurements, not that it ensures long-term stability, except if the lab temperature can be kept constant to a small fraction of a degree over a long time-period and the fibre is not stressed or strained. We have demonstrated a laser that is capable of detecting temperature variations of 0.003 °C over 9 cm fibre segments, with potential application in distributed sensing. The measurement is experimental proof that the laser is sensitive to minute temperature variations. By immersing the fibre in water inside a polystyrene ice box it was possible to carry out hourly-long runs of experiments without a measurable wavelength drift.

We do not have the means to keep the temperature of the lab and of the laser stable in a long-term stability test except for keeping the room air conditioning at the same setting (22 °C \pm 1 °C) and protecting the fibre in the cavity against air currents. Nevertheless, the wavelength drift observed is limited. Following the referee's request, we performed a wavelength stability test over a day's work in the lab, once the temperature reached equilibrium. We immersed only the region of the fibre addressed

in a water container. The figure below illustrates a long-term measurement of the laser stability. We stress that the horizontal scale is 1 pm per minor tick, which corresponds roughly to a temperature shift 0.1 °C. Over the 10 h stability test, the wavelength drifts by less than 0.0005 nm.

We remark that it is difficult to package a fibre laser to make it environment insensitive, and at the same time use the same fibre for sensing. The referee is right in that if this fibre laser is to be sold commercially as a wavelength stable fibre source, it will probably need active temperature stabilization in its package, but that probably precludes using the laser in a sensor system as demonstrated in our paper. This packaging is beyond the scope of the present study.

We thank the reviewer for this comment and add the figure to the supplementary material.

Long term stability test of the laser wavelength, which here drifts by less than 0.5 pm over 10 hours.

5. In simulating the Rayleigh scattering spectra of optical fibre sections with different lengths, how is “d = 5 mm” chosen so as to correspond to the experiment?

We thank the referee for the new query. There are many methods in the literature that simulate Rayleigh scattering in single-mode fibres. However, only a few of these address the problem in the spectral domain [1,2 below]. Our simulation was based in the method developed in ref. [1], where authors show that refractive index fluctuations in a spatial-scale much smaller than the wavelength of light are not enough to explain spectral fluctuations (and hence, cannot explain the interference pattern observed in phase-OTDR systems). It was argued there that a refractive index fluctuation on a larger spatial-scale (10^{-7} index variation every $d \approx 1$ cm) must be present to support spectral fluctuations. These larger spatial-scale index fluctuations are described as an artifact from the fabrication process, possibly coming from the imprecision in the feedback control loop required to maintain industry specifications (e.g. cladding diameter of 125 μm with acceptable deviation of 0.5 μm). Since fibres in the km-scale were simulated in ref. [1], experimental results could be well reproduced through simulations by assuming a spatial-scale variation of the refractive index of $d = 1$ cm. In the present paper, as we investigate the Rayleigh backscattering of much shorter fibre sections (< 2 m), the impact of a poor choice of d is much more relevant. We verified that $d = 1$ cm is not sufficient to represent the spectra obtained

experimentally. For instance, the mean peak width obtained when simulating a 0.5 m-long fibre with $d = 1$ cm was ≈ 160 MHz, i.e., slightly shifted from the 200 MHz observed experimentally. By reducing d to 5 mm, simulation results better reproduce the experimental measurements. Setting d to values lower than 5 mm (we simulated from 5 mm down to 0.1 mm) leads to the same statistical distribution obtained with $d = 5$ mm, but demanding higher computational power/longer simulation times. Values of d much smaller than 0.1 mm are unreasonable according to [1] since index fluctuations on a larger spatial scale are required to explain spectral fluctuations.

From a physical perspective, the exact value of d depends on the minutiae of the fabrication process of the fibre under test, and a small deviation between similar fibres is expected. For the single-mode fibre used in this work, the distribution of peak bandwidths was well represented through simulations when using $d = 5$ mm as shown in Fig. 2, which was obtained with an acceptable simulation time (< 10 min).

References:

- [1] P. Tovar, B. C. Lima and J. P. von der Weid, "Modelling Intensity Fluctuations of Rayleigh Backscattered Coherent Light in Single-Mode Fibres," in *Journal of Lightwave Technology*, vol. 40, no. 14, pp. 4765-4775, 15 July 2022, doi: 10.1109/JLT.2022.3164793.
- [2] Ushakov, N.A., Liokumovich, L.B. Comparison of Time and Frequency Approaches to Simulation of Signals of Optical Rayleigh Reflectometers. *Instrum Exp Tech* 66, 809–816 (2023). <https://doi.org/10.1134/S0020441223050147>

6. The novelty and significance of the proposed should be made more clear, especially, comparing with former works e.g., Refs [1] "Ultrafast Raman fibre Laser with Random Distributed Feedback," *Laser Photon. Rev.*, 12 (2018). [2] "Quasi mode-locking of coherent feedback random fibre laser," *Sci Rep*, 6, 39703 (2016), and [3] "Characterization of a FBG sensor interrogation system based on a mode-locked laser scheme," *Opt. Express* 25, 24650-24657 (2017).

We thank the referee for these comments, and address novelty and significance of our paper, and Ref. 1, Ref. 2 and Ref. 3 in the following.

The novelty of our work permeates the text, including the introduction. Our work is novel in that for the first-time transform-limited pulses were generated in an actively mode-locked random laser. We also discussed how perfect synchronous mode-locking is compatible with a random laser based on a distributed backscattering fibre. Our work is also novel in that we arguably demonstrate for the first time that the Rayleigh backscattering of centimetre-long telecom fibres is sufficient to achieve laser action. We also demonstrated for the first time that the coloured Rayleigh backscattering in a fibre can be used for self-adjusted spectral filtering in a laser cavity, which leads to transform-limited pulse generation irrespective of the injected pulse duration (within the limits of the study). We further demonstrate that this random laser can be used as a source of transform-limited pulses that can be electronically adjusted in pulse duration by nearly two orders of magnitude (0.34-20 ns) and a repetition rate that can be increased by 70%, from 0.714 to 1.22 MHz.

The significance of our work, besides all the new physics described here, is illustrated in a fully distributed temperature sensor operating inside a laser cavity with an open-ended fibre. The high spectral efficiency of the pulses leads to a remarkable spatial and temperature resolution of 9.0 cm and 3.3×10^{-3} K, respectively, in a fibre that is off-the-shelf, commercial, without treatment and without fabricated components such as FBGs. The precision demonstrated is sufficient for several

applications in the field, such as monitoring water leaks in dams. These aspects of novelty and significance are highlighted throughout the text.

[Ref 1]

As already discussed in the first round of reviews, the laser from [Ref 1] is quite different from ours. Ref 1 exploits Raman gain from repetitive pulse pumping, where a set of random cavities are selected adaptively and synchronize with the pump pulse train. In Ref 1, no definite cavity length exists in the random distributed feedback laser. A stable picosecond pulsed random laser is generated at 1120 nm. The spectrum is several nanometres wide, and the pulses are not transform-limited. The laser in Ref. 1 operates with several pulses circulating in the cavity at once. There, any picosecond laser pulse meets a cascade of pump pulses and reciprocally, under every pump pulse there are contributions from many SMF sections.

In contrast, our gain media are a SOA and EDFAs at 1.5 μm , the pulses are typically nanoseconds long and spectrally four orders of magnitude narrower than in Ref 1. In our laser, we have synchronous mode-locking, generating transform-limited pulses. In our work, the roundtrip time of any pulse chosen is well defined, and we have a single pulse circulating in the laser cavity. The laser pulse is repetitively amplified at a single roundtrip time associated with that unique pulse. In our work, we have a single backreflector for every pulse that we generate. In contrast to Ref. 1, we can adjust the pulse duration and repetition frequency of pulses electronically with the turn of a knob. Our laser is open-ended and the fibre is perfectly suited for distributed high-precision sensing, as we demonstrated in our manuscript.

[Ref 2]

In the work reported in [Ref 2], which we now add to the citation list as [Ref. 23], Q-switched trains of pulses were generated containing quasi mode-locked pulses and sub-pulses with multiple repetition frequencies. The mode-locked pulses correspond to several cavity lengths, and this was attributed to the presence of sub-reflections within their random FBG. They combined their strong random FBG (reflectivity greater than 20 dB) with a passive saturable absorber to provide a novel mechanism of generating collective resonances over low and high frequency regimes. They also found that the quasi-mode-locking of pulses/sub-pulses in their laser has a few repetition rates in different Q-switched envelopes.

In contrast, our laser is truly mode-locked. It does not use a saturable absorber. It does not Q-switch. The laser has a single cavity length for every repetition frequency set in the pulse generator. It does not have a strong random FBG, rather it explores the distributed Rayleigh scattering in any piece of fibre that has a reflectivity of typically -80 to -100 dB. Therefore, our backreflector can be easily addressable, and set to nearly any fibre location, while the random FBG explored in Ref 2 is only inscribed at a certain position in the fibre. In addition, our laser does not have sub-cavities, sub-reflections or sub-pulses. It is a distributed fibre sensor.

[Ref 3]

The work described in Ref. [3], cited by us as Ref. [46] in our previous text, describes a lasing interrogator system capable of reading the wavelength of FBGs inside a laser cavity. The cavity design has similarities to the one used here (and thus the citation). There, a CW SOA provides gain and an electrooptical modulator generating a sinusoidal modulation is frequency tuned for resonance at every FBG position.

In contrast to Ref 3, our work describes a Random laser. It is based on a cavity where one mirror is missing, and Rayleigh backscattering provides minute optical feedback. Ref. 46 addresses a conventional laser cavity with mirrors, one of which is a high reflectivity FBG (50% or more). Ref. 46 addresses FBG interrogation in a conventional laser, not distributed scattering. The possibility of doing distributed sensing is also lacking in Ref. 46. The fundamental question of whether it is possible to mode-lock a laser in a cavity with ill-defined modes does not arise in Ref 46. Although Ref 46 is an interesting piece of work, it is fundamentally different than the work described here, and more akin to Morey's paper from 1993 below [A. D. Kersey and W. W. Morey, "Multiplexed Bragg grating fibre-laser strain sensor system with mode-locked interrogation," *Electron. Lett.* **29**(1), 112–114 (1993).]

We are grateful to the reviewer for raising these points and bringing Ref 2 to our attention, which was missing from our reference list. We now incorporate an extra sentence citing [Refs. 1 and 2] above, to highlight in general terms some distinctions of our work and those publications. We add this sentence to the Discussion section of our manuscript, and make it very concise to avoid saying poorly what those authors wrote more clearly in their own papers. We do not include these considerations earlier in the text because a comparison between our work and those papers is difficult without describing our findings first.

Reviewer #2 (Remarks to the Author):

I believe that the authors have addressed the issues raised by the two referees in a satisfactory way. In particular, they clarified to me that the random scattering for the mode generation and the mode locking are two separate mechanisms. Although there are several reports demonstrating how mode locking in random lasers may rely entirely on randomness (see refs 13 and 14 of their manuscript), in their rebuttal they explained how in this case Mode locking is not correlated to randomness, but it is only possible to the coherent feedback provided by the round-trip cavity path. This is a fundamental distinction, that doesn't lead to classify their approach as "not random", but help separate their proof of concept laser architecture from the other.

I think that the paper is significantly improved and its now acceptable for publication in nature communications.

Appendix:

Data with time and spectral domain measurements of laser pulses. The time-bandwidth product for 6 pulse repetition frequencies over the range studied 0.7 – 1.2 MHz is calculated and compared. The pulse duration chosen was approximately 2 ns and the pulse profile was approximately square. Consistently, the spectra of the pulses (which was collected in a single sweep without averaging) was fitted to a function Sinc^2 . Limitations in the available electronics prevented obtaining entirely flat current pulses. The average time-bandwidth product measured was 0.83 instead of the 0.89 expected in the ideal case of perfectly square pulses and perfect Sinc^2 spectra (~7% difference).

F = 1203 kHz

$\Delta\lambda$ (FWHM) = 3.12 pm; $\Delta\nu = 3.99 \times 10^8$ Hz; ΔT (FWHM) = 2.1 ns; **$\Delta\nu \Delta T = 0.838$**

F = 1105 kHz

$\Delta\lambda$ (FWHM) = 2.91 pm; $\Delta\nu = 3.72 \times 10^8$ Hz; ΔT (FWHM) = 2.22 ns; **$\Delta\nu \Delta T = 0.827$**

F = 1003 kHz

$\Delta\lambda$ (FWHM) = 2.64 pm; $\Delta\nu$ = 3.38×10^8 Hz; ΔT (FWHM) = 2.0 ns; $\Delta\nu \Delta T = \mathbf{0.676}$

F = 903 kHz

$\Delta\lambda$ (FWHM) = 4.0 pm; $\Delta\nu$ = 5.12×10^8 Hz; ΔT (FWHM) = 1.73 ns; $\Delta\nu \Delta T = \mathbf{0.885}$

F = 805 kHz

$\Delta\lambda$ (FWHM) = 3.5 pm; $\Delta\nu$ = 4.48×10^8 Hz; ΔT (FWHM) = 1.83 ns; $\Delta\nu \Delta T = \mathbf{0.819}$

F = 735 kHz

$\Delta\lambda$ (FWHM) = 3.87 pm; $\Delta\nu$ = 4.95×10^8 Hz; ΔT (FWHM) = 1.85 ns; **$\Delta\nu \Delta T = 0.916$**

REVIEWERS' COMMENTS

Reviewer #1 (Remarks to the Author):

Thanks for the authors reply. I have no other comment, and recommend to accept the manuscript.